# Altered composition and functional profile of high-density lipoprotein in leprosy patients

Robertha Mariana R. Lemes[1], Carlos Adriano de M. e Silva[2], Maria Ângela de M. Marques[2], Georgia C. Atella[3], José Augusto da C. Nery[4], Maria Renata S. Nogueira[5], Patricia S. Rosa[5], Cléverson T. Soares[5], Prithwiraj De[2], Delphi Chatterjee[2], Maria Cristina V. Pessolani[1☯], Cristiana S. de Macedo[1,6☯]*

1 Laboratório de Microbiologia Celular, Instituto Oswaldo Cruz, Fundação Oswaldo Cruz, Rio de Janeiro, Rio de Janeiro, Brazil, 2 Mycobacteria Research Laboratories, Department of Microbiology, Immunology and Pathology, Colorado State University, Fort Collins, Colorado, United States of America, 3 Laboratório de Bioquímica de Lipídeos e Lipoproteínas, Instituto de Bioquímica Médica, Universidade Federal do Rio de Janeiro, Rio de Janeiro, Rio de Janeiro, Brazil, 4 Ambulatório Souza Araújo, Instituto Oswaldo Cruz, Fundação Oswaldo Cruz, Rio de Janeiro, Rio de Janeiro, Brazil, 5 Instituto Lauro de Souza Lima, Bauru, São Paulo, Brazil, 6 Centro de Desenvolvimento Tecnológico em Saúde, Fundação Oswaldo Cruz, Rio de Janeiro, Rio de Janeiro, Brazil

☯ These authors contributed equally to this work.
* cristiana.macedo@cdts.fiocruz.br

**Data Availability Statement:** All relevant data are within the manuscript and its Supporting Information files.

## Abstract

The changes in host lipid metabolism during leprosy have been correlated to fatty acid alterations in serum and with high-density lipoprotein (HDL) dysfunctionality. This is most evident in multibacillary leprosy patients (Mb), who present an accumulation of host lipids in Schwann cells and macrophages. This accumulation in host peripheral tissues should be withdrawn by HDL, but it is unclear why this lipoprotein from Mb patients loses this function. To investigate HDL metabolism changes during the course of leprosy, HDL composition and functionality of Mb, Pb patients (paucibacillary) pre- or post-multidrug therapy (MDT) and HC (healthy controls) were analyzed. Mb pre-MDT patients presented lower levels of HDL-cholesterol compared to HC. Moreover, Ultra Performance Liquid Chromatography-Mass Spectrometry lipidomics of HDL showed an altered lipid profile of Mb pre-MDT compared to HC and Pb patients. In functional tests, HDL from Mb pre-MDT patients showed impaired anti-inflammatory and anti-oxidative stress activities and a lower cholesterol acceptor capacity compared to other groups. Mb pre-MDT showed lower concentrations of ApoA-I (apolipoprotein A-I), the major HDL protein, when compared to HC, with a post-MDT recovery. Changes in ApoA-I expression could also be observed in *M. leprae*-infected hepatic cells. The presence of bacilli in the liver of a Mb patient, along with cell damage, indicated hepatic involvement during leprosy, which may reflect on ApoA-I expression. Together, altered compositional and functional profiles observed on HDL of Mb patients can explain metabolic and physiological changes observed in Mb leprosy, contributing to a better understanding of its pathogenesis.

**Funding:** This work was funded by grants from Heiser Program for Research in Leprosy and the Support for Strategic Research in Health (PAPES VI FIOCRUZ, National Council for Scientific and Technological Development-CNPq, Brasília-DF, Brazil), RMRL was supported by a doctoral fellowship from the Instituto Oswaldo Cruz (IOC-Brazil), PDSE program from Coordination of Superior Level Staff Improvement (CAPES, Brasília-DF,Brazil), and a postdoctoral fellowship from National Institute for Science and Technology-Innovation on Diseases of Neglected Populations (INCT-IDPN, CNPq). CAMS was supported by a postdoctoral fellowship from Science Without Borders program-CAPES/Brazil. The funders had no role in study design, data collection and analysis, decision to publish, or preparation of the manuscript.

**Competing interests:** The authors have declared that no competing interests exist.

## Author summary

Leprosy is a chronic disease caused by *Mycobacterium leprae*, which causes lesions on the skin and peripheral nerves. Some patients do not present an efficient immune response and have a disseminated infection (multibacillary, Mb). Mb patients have lipid accumulation in infected tissues that is important for microorganism survival. High-density lipoprotein (HDL) is composed of proteins and lipids and is produced in the liver. It removes excess of lipids from peripheral tissues and presents anti-inflammatory activity; however, these activities are not being properly performed in leprosy. To understand more about HDL metabolism on leprosy, the chemical composition and functionality of HDL from leprosy patients were analyzed before and after treatment with antibiotics (multidrug therapy, MDT). It was observed that HDL has an altered lipid composition in Mb patients before MDT, which may lead to an impairment of its functions. Apolipoprotein A-I (ApoA-I), the main HDL protein, seems to be highly affected during infection. These functions can be slightly recovered after MDT, but not in the levels of healthy individuals. Our data open new perspectives to elucidate the modulation of lipid metabolism in leprosy and consequently to prevent disease complications.

## Introduction

Leprosy is a human disease characterized by a spectrum of clinical manifestations that occurs due to a wide range of immune responses against *Mycobacterium leprae*, the causative agent of the disease. This illness mainly affects the skin and peripheral nerves and can lead to deformities and disabilities. Despite the efficacy of multidrug therapy (MDT), leprosy still affects approximately 200,000 people globally [1]. To favor diagnosis and treatment, the World Health Organization (WHO) sorted leprosy patients into two groups based on the number of lesions and their bacillary load: paucibacillary (Pb) and multibacillary (Mb) [2]. Pb patients display a strong specific cellular immune response against *M. leprae*, presenting few skin lesions and negative bacillary index (BI) in skin or lymph smear while Mb patients present systemic manifestations and a positive BI with more than five skin lesions since the immune system is unable to control the multiplication of the bacilli [3–5].

Lesions of Mb patients usually exhibit lipid-loaded macrophages, known as foam or Virchow cells. Previous works from our group demonstrated that Virchow cells found in dermal lesions of Mb patients are highly positive for adipose differentiation-related protein (ADRP), a classical marker for the cellular organelles called lipid droplets [6]. These data indicated that at least part of the foamy phenotype in Virchow cells is due to the accumulation of these organelles. Our group showed in nerve biopsies of Mb patients and *in vitro* cultures infected with *M. leprae* that, besides macrophages, Schwann cells displayed a great number of lipid droplets [6, 7]. Furthermore, the accumulation of lipid droplets is induced by *M. leprae*, favoring its own survival [7, 8]. The majority of lipids present in these organelles are derived from the human host, which is in agreement with the work of Cruz et al. [9] that revealed a higher expression of genes involved with the metabolism and synthesis of lipids in skin lesions of Mb subjects [6–8, 10, 11]. It was also shown that high bacillary loads of *M. leprae* increase the uptake of LDL-cholesterol (low-density lipoprotein) and induce the biosynthesis of cholesterol via the upregulation of the expression of LDL receptor and hydroxymethylglutaryl coenzyme A (HMG-CoA) reductase, respectively [8, 12, 13]. Additionally, Cruz et al. [9] findings indicated that high-density lipoprotein (HDL) from Mb patients is not functional. This possibly favors the accumulation of oxidized phospholipids in Virchow cells, since the main HDL function is the

reverse transport of cholesterol, phospholipids and non-esterified fatty acids from peripheral tissues to the liver [14–16], being able to reverse the formation of foam cells and reducing the volume of atherosclerotic plaques [17]. Together, these results suggest that the dysfunctionality of HDL is related to the formation of Virchow cells. However, it is unclear why HDL from Mb patients loses its function.

Changes in the functional and compositional profile of HDL particles are mainly caused by apolipoprotein A-I (ApoA-I) deficit and were shown to be linked to the inflammatory response in several contexts [18–21], including systemic inflammation triggered by infections, together with hepatic involvement [22–24]. In the present study, we propose that cellular lipid accumulation observed in the systemic *M. leprae* infection of Mb patients may also be due to HDL dysfunctionality, which would be related to altered ApoA-I levels and lipid composition of HDL particles. To test this hypothesis, the composition and biological activities of HDL isolated from Mb patients before (pre-MDT) and post-MDT were compared. Both healthy volunteers and Pb patients were included as controls. The capacity of *M. leprae* to modulate the biosynthesis of ApoA-I in hepatic cells infected *in vitro* was also analyzed.

## Methods

### Patients and healthy volunteers

The present study comprised 39 voluntary participants (Table 1): 12 healthy controls (HC), 8 leprosy multibacillary patients before MDT (Mb pre-MDT), 6 leprosy multibacillary patients after MDT (Mb post-MDT), 6 leprosy paucibacillary patients before MDT (Pb pre-MDT) and 7 leprosy paucibacillary patients after MDT (Pb post-MDT). Leprosy patients were classified according to WHO criteria, where all those with positive bacterial index (BI>1) were classified as Mb leprosy, while those with negative BI were assigned as Pb [25]. The clinical specimens of the Pb pre-MDT and Mb pre-MDT groups were collected before the beginning of treatment, while the clinical specimens of Pb post-MDT and the Mb post-MDT were collected immediately after the end of treatment. Pb post-MDT patients correspond to those who were treated with 6 doses, and Mb post-MDT patients, with 12 doses of MDT, according to the criteria established by WHO [26, 27]. Both Mb and Pb patients included in pre-MDT or post-MDT groups are different individuals. All participants were non-smokers and leprosy patients had no diagnosed comorbidities. Leprosy patients were recruited from "Souza Araújo" Leprosy Outpatient Unit (Oswaldo Cruz Foundation, Rio de Janeiro-RJ, Brazil). Healthy controls, all residents in the city of Rio de Janeiro (State of Rio de Janeiro, Brazil), were selected according to the similarity of age and gender patient's cohort and weren't contacts of leprosy patients.

### Sample collection

Without fasting, blood samples from Pb, Mb leprosy patients, and HC were collected in tubes containing sodium citrate. Part of the plasma recovered after centrifugation was stocked at 4°C for cholesterol, HDL-cholesterol, ApoA-I and PON-1 measurements. The other part of fresh plasma was immediately used for lipoprotein separation by HPLC.

### Ethics statement

The use of plasma samples was approved by the FIOCRUZ Human Ethics Committee (number 504/09). All participants, including parents of minors, provided informed written consent.

**Table 1. Baseline characteristics of the study population and levels of cholesterol, ApoA-I and PON1.**

| | Group | HC | Mb pre-MDT | Mb post-MDT | Pb pre-MDT | Pb post-MDT |
|---|---|---|---|---|---|---|
| | Participants (n) | 12 | 8 | 6 | 6 | 7 |
| | Female | 4 | 1 | 2 | 4 | 3 |
| | Male | 8 | 7 | 4 | 2 | 4 |
| | Age[a] median (min-max) | 41 (20–66) | 40 (17–51) | 33,5 (26–50) | 44 (19–69) | 41 (15–49) |
| | BI[b] median (min-max) | - | 4,87(2,75–5,75) | 5 (2,75–5,75) | 0 | 0 |
| Plasma levels median (min-max) | Total-cholesterol[c] | 181 (161–221) n = 12 | 152 (127–221) n = 8 | 172 (138–186) n = 6 | 166 (110–286) n = 6 | 163 (143–224) n = 7 |
| | HDL-cholesterol[c] | 62 (55–74) n = 12 | 31,5 (21–39) n = 8 | 51,5 (47–66) n = 6 | 55,5 (25–70) n = 6 | 48 (45–69) n = 7 |
| | ApoA-I[c] | 184 (137–257) n = 12 | 80,8 (44–83) n = 5 | 139 (97–170) n = 6 | 134 (70–205) n = 6 | 114 (79–275) n = 6 |
| | PON1[c] | 37 (24–57) n = 7 | 48,1 (44,4–51,8) n = 6 | 16,5 (27,8–63,1) n = 5 | 44,9 (37,3–64,5) n = 5 | 43,3 (37,2–52,6) n = 7 |

Groups included in this study: HC: Healthy Control, Mb: Multibacillary; Pb: Paucibacillary.

[a]Age in years.

[b]BI: baciloscopic index.

[c]Concentration in mg/dL.

ApoA-I: Apolipoprotein A-I.

HDL: High-density lipoprotein.

PON1: Paraoxonase 1.

MDT: Multidrug therapy.

### Liver autopsy tissue

The liver autopsy was performed on a male patient, 78 years old, who was diagnosed with leprosy and classified as Mb. The liver autopsy fragment was preserved since 1982 and donated by Lauro de Souza Lima Institute (ILSL), Bauru-SP, Brazil.

### Plasma levels of total cholesterol, HDL-cholesterol, ApoA-I and PON1

Total cholesterol and HDL-cholesterol were measured by 'Monoreagent Cholesterol' (Bioclin, Brazil) and 'Direct HDL-cholesterol' (Doles, Brazil) kits, respectively, according to the manufacturer's instructions. ApoA-I and PON1 were measured by ApoA-I antibody-G2 EIA kit (Abcam, MA, USA) and Human Total PON1 EIA Kit (R&D Systems, MN, USA), according to each manufacturer.

### HDL purification

Fresh plasma from HC, Pb and Mb patients (200 μL) were injected in two Superose 6HR columns (GE Healthcare, USA), positioned in tandem on a HPLC (high-performance liquid chromatography, LC-10AS, Shimadzu, Japan) with PBS and EDTA (0.001 mM) as mobile phase, in a 0.5 mL/min flow [28, 29]. Protein estimation was performed by SPD-10AUV (Shimadzu) at 280 nm and fractions were collected every minute. The purity and integrity of HDL were assessed by loading 20 μL of each HPLC fraction into 7.5% native PAGE (polyacrylamide gel electrophoresis) (adapted from [30]). To identify HDL on HPLC fractions, major constituent proteins and the cholesterol profile were accessed by 15% SDS-PAGE (stained by Coomassie Blue) and 'Cholesterol Monoreagent' kit, respectively. The presence of ApoA-I was

confirmed loading equal amounts (20 μl) of each HPLC fraction rich in HDL into 15% SDS-PAGE, followed by immunoblot through anti-ApoA-I primary antibody (Abcam) and secondary anti-mouse horseradish peroxidase-conjugated antibody (Merck). Protein bands were visualized by ECL reagent (GE Healthcare).

## Live *M. leprae* and cell cultures

*M. leprae* was purified from athymic *nu/nu* mice infected footpads, as described [31], and donated by ILSL. All cell lines were incubated under standard conditions (5% $CO_2$, 37˚C). HCAEC (Human Coronary Artery Endothelial Cell, donated by Dr. Karyn Hamilton from CSU, USA) were grown in ECBM-2 medium supplemented as recommended by the manufacturer (Lonza, Basel, Switzerland). THP-1 (human leukemic monocyte TIB-202; ATCC, USA) were grown in RPMI medium supplemented with FBS (fetal bovine serum, 10%) and differentiated with PMA (phorbol 12-myristate 13-acetate, 50 ng/mL, Merck) for 24 hours. HepG2 (Human Hepatoma Cells, ATCC) were grown in DMEM medium supplemented with FBS (10%).

## HDL anti-inflammatory activity assays

**MCP-1 measurement:** Cultured HCAEC cells were incubated in the presence or absence of HDL (50 μg/mL from Pb, Mb or HC) for 24 hours in ECBM-2 medium without supplementation. MCP-1 was measured in the supernatant by EIA (Human MCP-1 Kit; R&D Systems). **IL-6 measurement:** For IL-6 (interleukin 6) quantification, HCAEC cells were firstly stimulated with HDL (50 μg/mL from Pb, Mb or HC) for 2 hours and then treated or not with LPS (lipopolysaccharide, 0.5 μg/mL) for more 24 hours. IL-6 was measured on the supernatant by Human IL-6 EIA Kit (R&D Systems).

## HDL antioxidant activity assay

Cultured HCAEC cells were pre-stimulated with HDL (50 μg/mL from Pb, Mb or HC) for 10 min in ECBM-2 medium without supplementation. $H_2O_2$ (hydrogen peroxide) at 25 μM (Merck) was added in each condition and incubated for one additional hour. Untreated controls were performed in parallel. After incubation, all conditions received CellROX Deep Red Assay Kit (Thermo Fisher), following the manufacturer's guidelines, and fluorescence of individual cells was assessed using FACSDIVA (BD Bioscience, Germany). Results were analyzed by FlowJo V10 (BD Bioscience).

## Cholesterol efflux activity assay

To measure cholesterol efflux by HDL, macrophages differentiated from THP-1 cells, as described, were firstly incubated with RPMI medium and FBS (5%) contain [4-$^{14}$C]cholesterol (0.1 μCi/mL; American Radiolabeled Chemicals, USA) during 48 hours. Subsequently, supernatants were removed, cells rinsed with PBS, followed by more 18 hours of incubation with BSA (bovine serum albumin 2%, diluted in RPMI medium). At the end, cells were exposed to HDL (50 μg/mL from Pb, Mb or HC) for six hours. Supernatants and cell lysates (prepared using RIPA buffer) were separately added to scintillation liquid (Optima Gold Plus, Perkin Elmer, USA). Radioactivity was measured using a scintillation counter (LS 600, Beckman Coulter, USA). Results were expressed in DPM (disintegrations per minute) and calculated as:

$$\% \text{ Cholesterol Efflux} = 100 \text{x} \frac{\text{supernant DPM}}{\text{cell lysated DPM} + \text{supernant DPM}}$$

## Histopathology of the liver autopsy

Liver autopsy fragment was fixed in 10% formalin for 48 hours, dehydrated through a series of graded ethanol baths, infiltrated and embedded into paraffin blocks and preserved since 1982. Recently, 3 μm sections were obtained and stained with H&E (hematoxylin and eosin stain) [32] or Fite-Faraco [33]. Histopathological analysis was performed, and pictures were collected using an Axiophot 2 photomicroscope (Carl Zeiss, Germany) using 400 and 1000x magnification.

## Infection of hepatic cells with *M. leprae*

Cultured HepG2 cells were incubated for 24 hours with live *M. leprae* at different multiplicities of infection (MOI; 1:1, 10:1, 25:1 and 50:1 bacilli/cell) at 5% $CO_2$ and 33°C. After incubation, cells were lysed with RIPA buffer, and protein concentration was determined by BCA kit (Thermo Fisher). Twenty μg of protein of each condition were loaded to a 15% SDS-PAGE and ApoA-I immunoblotting was performed. Data were normalized by anti-GAPDH (anti-glyceraldehyde 3-phosphate dehydrogenase, Santa Cruz, TX, USA). For immunofluorescence, HepG2 cells were incubated for 24 hours with PKH26-labelled-*M. leprae* (MOI 10:1) (green PKH26, Merck). After incubation, cells were fixed with 4% paraformaldehyde, nuclei stained by DAPI (4',6-diamidino-2-phenylindole dihydrochloride, Thermo Fisher) and mounted with ProLong Gold Antifade (Thermo Fisher). Pictures were captured with a fluorescence microscope (Carl Zeiss) fitted up with a Plan Apo 100 objective.

## Statistical analysis of the assays

All groups presented non-normal distributions probably due to varied and low number of participants. Mann-Whitney test was used to compare two groups, while Kruskal-Wallis with the addition of Dunn's test was used for multiple comparisons. Values of *$p < 0.05$, **$p < 0.001$ and ***$p < 0.0001$ indicated statistically significant differences.

## Lipidomics

Equal amounts of protein (200 μg) from purified HDL samples were used for lipid extraction, as described [34], and submitted to UPLC-MS (ultra-performance liquid chromatography-mass spectrometry) to compare the lipidomic profile of HDL purified from leprosy patients and HC. Data were obtained in positive ionization mode with a Xevo G2 Q-TOF MS (Waters, MA, USA) under the command of the MassLynx v4.1 (Waters). **UPLC Injection**. Lipid samples were separated by reversed-phase chromatography on a C8 column (Waters ACQUITY UPLC BEH C8; 1.7 μm particle size, 1.06 x 100 mm) using solvent A (89% water, 5% acetonitrile, 5% isopropanol, 1% 500 mM ammonium acetate) and solvent B (49.5% acetonitrile, 49.5% isopropanol, 1% 500 mM ammonium formate). Each sample (1 μL) was injected in 100% of solvent A for 0.1 min. Then, solvent B (40%) was applied during 0.9 min, followed by 10 min of linear gradient of solvent B (100%), which was held by 3 min and then returned to starting conditions over 0.1 min, and then re-equilibrated for 5.9 min (20 min of total run time). Flow rate (140 μL/min) and column temperature (50°C) were held constant throughout the experiments. Mass spectral data were collected in centroid mode and in MSE (mean squared error) mode as described [35]. Electrospray ionization was achieved using a capillary voltage of 3 kV. Other MS parameters used: gas temperature, 350°C; drying gas flow rate, 800 L/min; sample cone, 30 V; extraction cone, 2.0 V; and 130°C of source temperature.

## Processing and statistical analysis of UPLC-MS data

The converted raw LC-MS files (supplemental methods) were processed for molecular feature (MFs, UPLC-MS signals defined by *m/z* and retention time values) detection, retention-time correction and chromatogram alignment through XCMS [36] in R software as described [35]. The raw intensity value of each MF was normalized by the quantile method [37]. After normalization, the MFs were grouped into spectra based on the co-elution and covariance across the full dataset by RAMClustR [35]. Through this approach, a compound group (named "compound") is assigned to spectra that include adducts, isotopes, and monoisotopic mass. The intensity for each "compound" represents the spectral intensity, which was obtained through a weighted mean function of the MFs intensities that were clustered in a "compound". The spectral intensity of each "compound" from a biological sample was summarized by the average of duplicate values, converted to $\log_2$ and compared between the groups under study (HC, Mb pre-MDT, Mb post-MDT, Pb pre-MDT, and Pb post-MDT) through linear models approach with t-statistics having an empirical Bayes moderation, implemented in the R package limma [38, 39]. The *p* values were adjusted for false discovery rate [40]. The "compound" with log2 fold change ($\log_2$FC)$\geq$1.0 and $p<0.05$ were considered statistically significant and selected for further identification. Principal component analysis (PCA) was also conducted on scaled and centered spectral intensity values of all "compounds" through prcomp function in R.

## Putative identification of the "compounds" in the Human Metabolome (HMDB) and LIPID MAPS databases

Spectra built by RAMClustR analysis for each "compound" with $\log_2$FC$\geq$1.0 and $p<0.05$ were used to determine its putative identity in the HMDB [41] and LIPID MAPS [42]. For the search, the adducts $[M+H]^+$ and $[M+Na]^+$ were mainly used. However, for some other "compounds" different adducts like +ACN, +K and $NH_4$ were also considered. A mass error tolerance of +/- 20 ppm was used (see S3 Table).

# Results

## HDL-cholesterol is reduced in multibacillary leprosy patients

HDL-cholesterol levels are an important predictor of systemic quantitative alterations in this particle since cholesterol on free or esterified form accounts for 40 and 10% of the total HDL lipids, respectively [43]. Since cholesterol metabolism is modulated by *M. leprae* infection [6, 8, 12, 13], the cholesterol content of HDL from leprosy patients and HC was investigated.

Total cholesterol and HDL-cholesterol levels were enzymatically measured in plasma and compared among untreated Mb and Pb leprosy patients (pre-MDT Mb and pre-MDT Pb) and HC. Plasma samples were also collected from patients immediately after the conclusion of MDT (post-MDT Mb and post-MDT Pb) to more clearly link the potential changes found during *M. leprae* infection. Mb pre-MDT patients did not show a significant difference on total cholesterol levels when compared to HC (Fig 1A and Table 1). When analyzing HDL-cholesterol, the differences observed between Mb pre-MDT patients and HC were higher, suggesting that the slightly but not significant difference detected in total cholesterol results from differences in cholesterol content in HDL fraction. HDL from Mb pre-MDT patients showed about a 50% reduction in cholesterol content when compared to HC group (Fig 1B and Table 1). The observed HDL-cholesterol levels in Mb pre-MDT patients were 31.5 (21–39) mg/dL [median (min-max)] and in HC group were 62 (55–74) mg/dL [median(min-max)] with a $p<0.0001$(Table 1). Interestingly, HDL-cholesterol levels of Mb patients returned to normal, baseline levels closer to those observed in HC after MDT conclusion. On the other

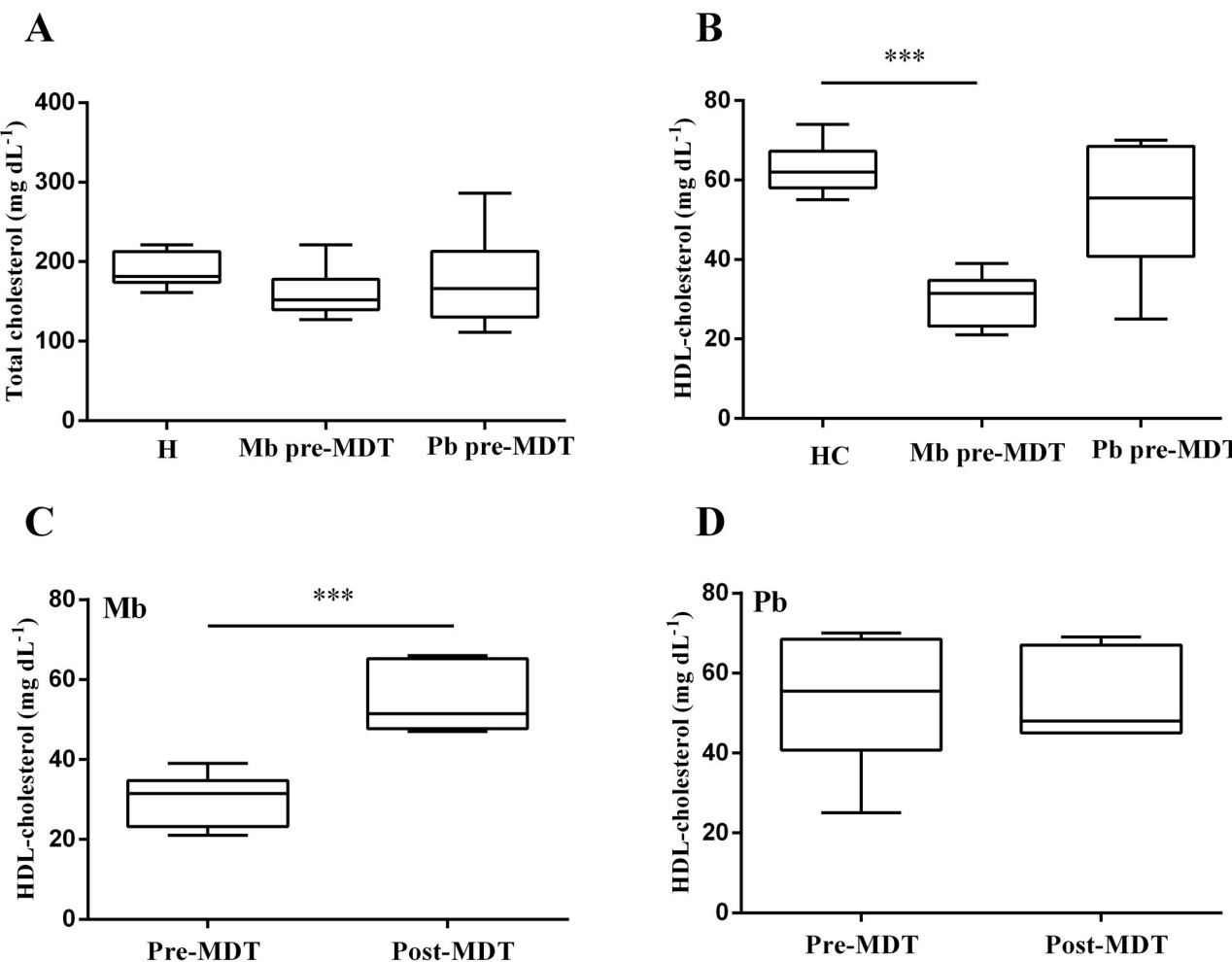

**Fig 1. Multibacillary leprosy patients display lower levels of HDL-cholesterol.** Box-plots represent total cholesterol (A) levels from HC (n = 12), Mb pre-MDT (n = 8), and Pb pre-MDT (n = 6). (B) HDL-cholesterol levels of HC (n = 12), Mb pre-MDT (n = 8), and Pb pre-MDT (n = 6). (C) HDL-cholesterol levels of Mb pre-MDT (n = 8) and Mb post-MDT (n = 6) (B). (D) HDL-cholesterol levels of Pb pre-MDT (n = 6) and Pb post-MDT (n = 7). Median and min-max values are indicated by lines. (A) and (B) HC, Mb pre-MDT and Pb-MDT comparisons were evaluated by Kruskal-Wallis non-parametric and Dunn's tests. (C) Mb and (D) Pb patients were evaluated with Mann-Whitney non-parametric test. ***$p \leq 0.0001$.

hand, no difference in HDL-cholesterol levels was observed between Pb patients (pre- and post-MDT).

## The lipidomic profile revealed an altered composition of HDL particles in multibacillary leprosy patients

Next, a lipidomics approach was used as an exploratory tool to assess differences in the composition of HDL obtained from HC and leprosy patients. Enriched HDL fractions were obtained from plasma by gel filtration chromatography as illustrated in Fig 2A. The chromatographic pattern based on protein estimation and cholesterol content of the fractions allowed to indicate that fractions 52–57 were enriched in LDL, and the fractions 68–84 contained HDL and albumin, according to the inversely proportional ratio of protein/cholesterol content [44]. Analysis by native-PAGE (Fig 2B) showed the presence of a major protein with a molecular weight corresponding to HDL size (150–360 kDa) in fractions 68–75. An additional analysis by SDS-PAGE (Fig 2C), followed by immunoblotting developed with an antibody against ApoA-I, the specific HDL protein (Fig 2D), confirmed that HDL is a major constituent of these fractions.

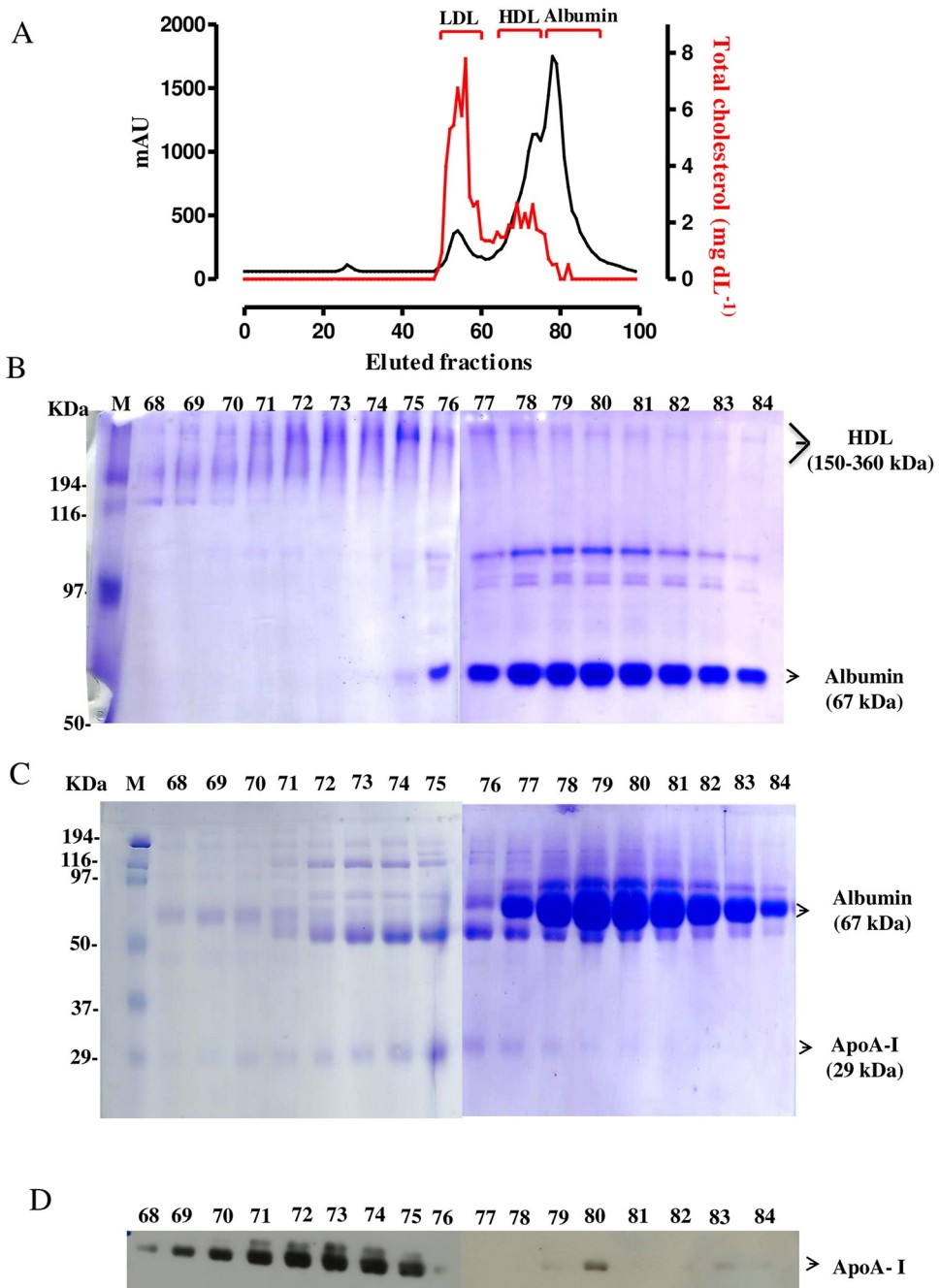

**Fig 2. Isolation of enriched HDL fractions.** (A) Plasma protein distribution on HPLC gel filtration chromatography fractions (black line, left axis, mAU, milli-absorbance unit, detected at 280 nm) and total cholesterol levels of fractions (red line, right axis, mg/dL), determined by enzymatic assay. (B) 7.5% native- and (C) 15% SDS-PAGE, and (D) ApoA-I immunoblotting profile of HPLC fractions (68–84). M: standard protein markers.

As described, the HDL separation needs to be performed in a carefully and longstanding manner, to prevent endotoxin contamination or protein oxidation. HPLC was preferred over ultracentrifugation because HDL fractions obtained by this method had higher purity, and better integrity compositional to perform lipidomics and functional assays [45, 46].

A total of 7012 molecular features were detected in positive ionization mode by XCMS and then grouped in 1260 "compounds" through ClustalR. A PCA was performed based on the intensity of the "compounds" and it was revealed that Pb pre-MDT, Pb post-MDT, and Mb post-MDT did not cluster away from HC (S1 Fig). Also, Pb pre-MDT group could not be distinguished from Pb post-MDT group (S2B Fig). On the other hand, Mb pre-MDT patients were separated from HC as well as from Pb pre-MDT patients (Fig 3). However, no clear separation was observed between Mb pre-MDT and Mb post-MDT groups. Statistical analysis revealed that only the comparisons HC vs Pb post-MDT (vs, versus), HC vs Mb pre-MDT and Pb pre-MDT vs Mb pre-MDT displayed few "compounds" with significantly altered intensities ($\log_2 FC \geq 1.0$ and $p < 0.05$) (see S1 and S2 Tables). The other comparisons did not show any "compounds" with $\log_2 FC \geq 1.0$ and $p < 0.05$. More specifically, the "compounds" C83, C200, C209, and C543 were significantly increased in Mb pre-MDT patients in comparison with HC (S2 Table), while C200, C209, C563 displayed a higher intensity in Mb pre-MDT when compared to Pb pre-MDT patients. Furthermore, C273, C608, C774, C835, C1062, and C1065 were lower in Mb pre-MDT group than in the Pb pre-MDT group. Unexpectedly, when HC with Pb post-MDT patients were compared, it was observed that the levels of C83, C102, C127,

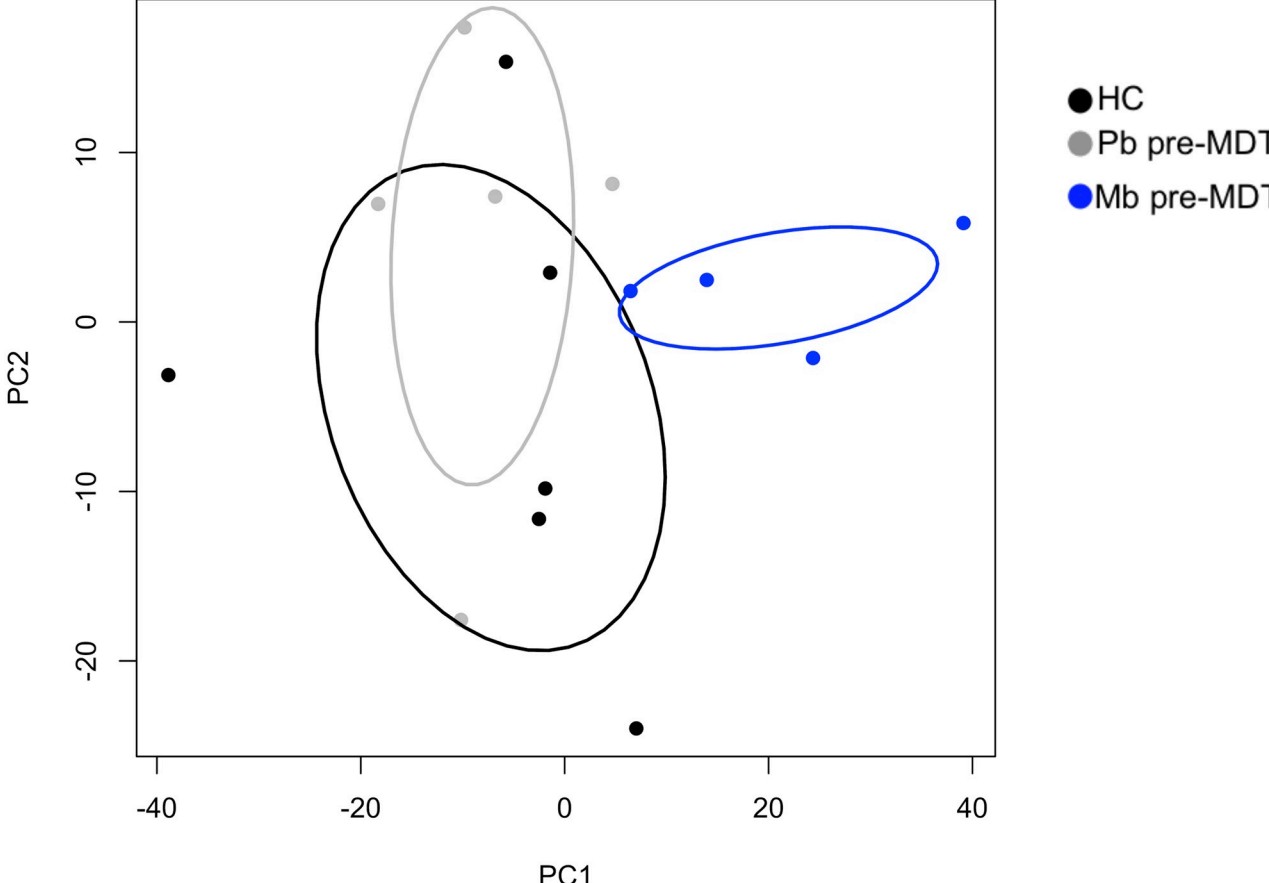

**Fig 3. The lipidomic profile revealed altered composition of HDL particles of multibacillary leprosy patients before treatment.** Raw UPLC-MS data collected in positive ionization mode were processed and analyzed by XCMS, followed by normalization through the quantile method. Subsequently, the molecular features (MFs) were grouped into spectra through RAMClustR. In this approach, each spectrum represents a "compound" with adducts, isotopes and monoisotopic mass. The intensities of 1260 "compounds" were used to perform a principal component analysis. HC (n = 6, black dots), Pb pre-MDT (n = 5, grey dots), Mb pre-MDT n = 4, blue dots).

C162, C543, and C999 augmented in Pb post-MDT patients, while the levels of C1020 were reduced. Although the lipid composition of HDL among leprosy patients could not be defined by using this lipidomic approach, our data indicate that HDL from Mb pre-MDT patients presents differences in its lipidomic profile in comparison with HC and Pb pre-MDT groups.

## HDL of multibacillary leprosy patients is dysfunctional

Since plasma concentrations of cholesterol were altered in Mb pre-MDT patients, the next step was to test the functional properties of HDL. Assaying the antioxidant function of HDL by a classic protocol using HCAEC cells, HDL from Mb pre-MDT showed a significantly lower capacity to decrease the propagation of reactive oxygen species (ROS) chain reactions induced by $H_2O_2$, unlike HDL from HC (Fig 4A). Interestingly, the antioxidant capacity of

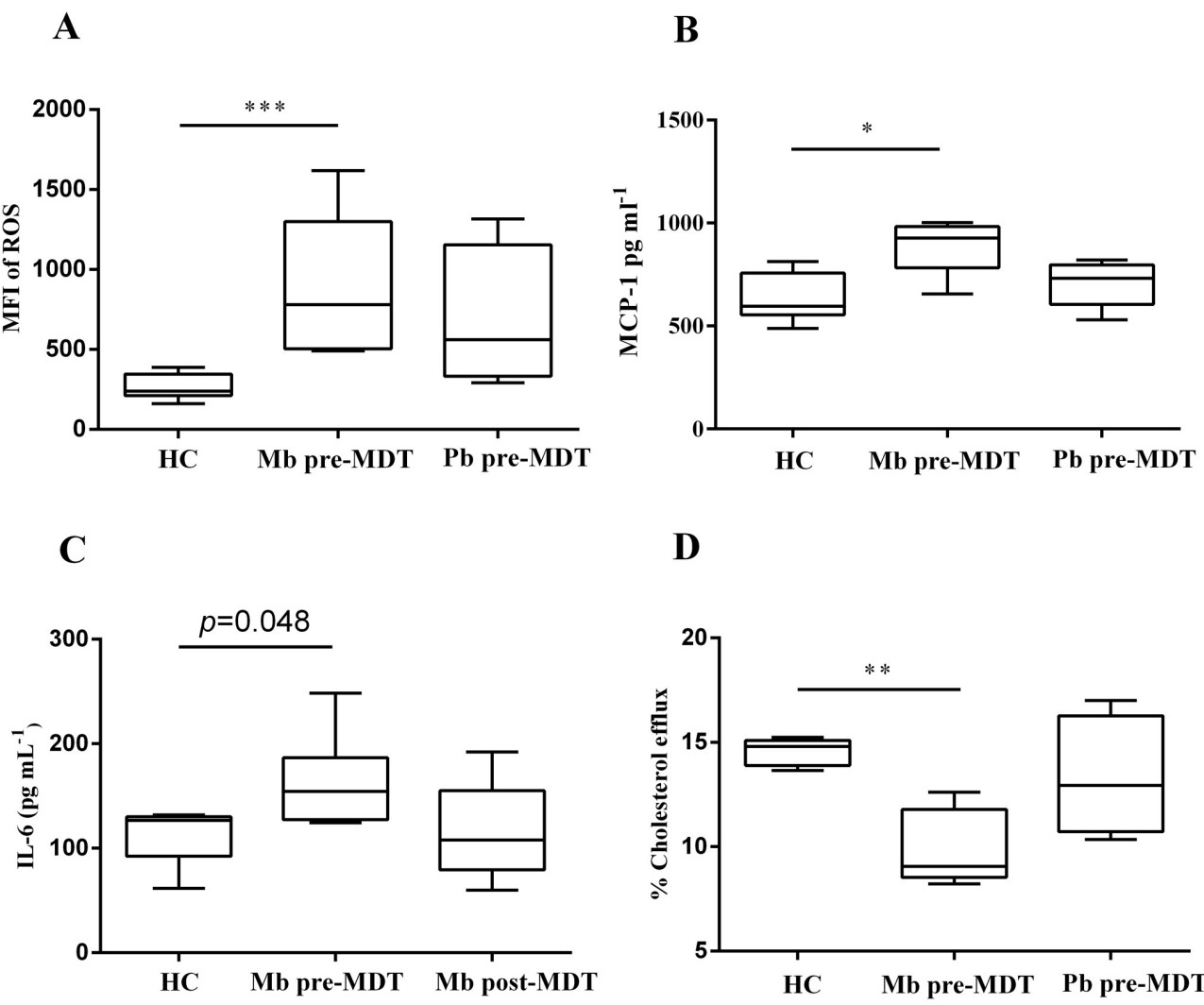

**Fig 4. HDL from multibacillary leprosy patients displays altered functions.** (A) Inhibition of ROS (reactive oxygen species) by HDL purified from leprosy patients and HC were measured by flow cytometry analysis. Data are expressed by the median fluorescence intensity (MFI). HC (n = 7), Mb pre-MDT (n = 5) and Pb pre-MDT (n = 5) (B) Expression of monocyte chemoattractant protein-1 (MCP-1) modulated by HDL from leprosy patients and HC in HCAEC cells. HC (n = 7), Mb pre-MDT (n = 5), and Pb pre-MDT (n = 6) (C) Interleukin-6 (IL-6) response to HDL of HC and leprosy patients expressed relative to response of HCAEC cells stimulated with LPS (0.5 μg/mL). HC (n = 5), Mb pre-MDT (n = 7), Mb post-MDT (n = 5). (D) Pb, Mb and HC HDL-cholesterol efflux from THP-1 differentiated macrophages. HC (n = 7), Mb pre-MDT (n = 5), and Pb pre-MDT (n = 4). Median and min-max values are indicated by lines. Group comparisons were evaluated using Kruskal-Wallis non-parametric and Dunn's test. $^*p \leq 0.05$ and $^{**}p \leq 0.001$.

HDL was at least partially improved after Mb and patients concluded the MDT regimen (Mb post-MDT), although no significant differences were observed (S3A Fig). HDL from Pb patients did not present significant differences when compared to healthy donors (Fig 4A).

The anti-inflammatory activity of HDL from leprosy patients was assessed by two methodologies. Firstly, the well-known capacity of HDL to decrease the basal expression levels of monocyte chemoattractant protein-1 (MCP-1) in endothelial cells was analyzed [21, 47] (Fig 4B). In contrast to HDL from HC, HDL from Mb pre-MDT patients was unable to significantly decrease the expression of MCP-1 in HCAEC cells, having almost 50% of HC's HDL performance. HDL from Mb post-MDT retained the same behavior, although a tendency to recover this function was observed (Fig 4B and S3B Fig). On a second inflammatory assay, HCAEC cells were previously sensitized with LPS (lipopolysaccharide), a TLR4 (Toll-like receptor 4) agonist, and IL-6 production in the presence of HDL was measured. The levels of IL-6 expression were significantly higher in the presence of HDL from Mb pre-MDT, in comparison with HDL from HC. In this assay, HDL from Mb post-MDT patients recovered partially the anti-inflammatory activity observed on HDL from HC (Fig 4C).

Finally, the reverse cholesterol transport activity, which is the major HDL function, was measured in enriched HDL fractions. HDL from Mb pre-MDT was markedly dysfunctional, presenting almost half-fold of activity compared to the HC group (Fig 4D). HDL from Mb post-MDT also showed diminished reverse transport activity, but a tendency to recover this function was observed (S3C Fig). No significant differences were observed between Pb patients and HC's HDL performance (Fig 4D).

## Plasma ApoA-I is highly reduced in multibacillary patients

Changes in the function of HDL particles are mainly caused by ApoA-I deficit. Then, as a next step, the plasma levels of ApoA-I were measured in leprosy patients. When measured by EIA (Fig 5A, Table 1), plasma ApoA-I levels were approximately two half-fold lower in Mb pre-MDT compared to plasma from HC. No statistically significant difference was observed between Pb pre-MDT and HC (Fig 5A). Of note, MDT treatment allowed the recovery of ApoA-I plasma levels in Mb (Mb post-MDT) patients (Fig 5B).

The plasma levels of paraoxonase 1 (PON1), an enzyme mainly associated with HDL, were measured by EIA (Fig 5B, Table 1). PON1 possesses expressive antioxidant and cholesterol efflux-stimulating activities. In plasma, it is exclusively associated with HDL after being expressed by the liver [48–51]. Although PON1 has an important role in the structure and functionality of HDL, no significant differences were observed on its expression between plasma samples of patients and HC.

## *M. leprae* decreases ApoA-I expression in infected hepatic cells *in vitro*

To gain further insight into the origin of dysfunctional activity of HDL in Mb patients, the next step focused on investigating if *M. leprae* can modulate the expression of ApoA-I in hepatic cells since the liver is responsible for ApoA-I biosynthesis and reports are indicating hepatic alterations in leprosy [52, 53]. Firstly, it was checked the hepatic involvement in a liver autopsy from one Mb patient. Photomicrographs of liver sections showed hepatic parenchyma with preserved architecture, vascular congestion (stars), foci of macro and microvesicular steatosis (black arrows), extended portal space with ductal proliferation (arrowheads), unobtrusive fibrosis and mononuclear inflammatory infiltrate containing lymphocytes and vacuolated macrophages (white arrows) (Fig 6A). These vacuolated macrophages presented AFB (granular acid-fast bacilli) in their interior (Virchow's cells, stars), which were found both in the inflammatory infiltrate (Fig 6B) and in hepatic sinusoids (Fig 6C). Taken those data together,

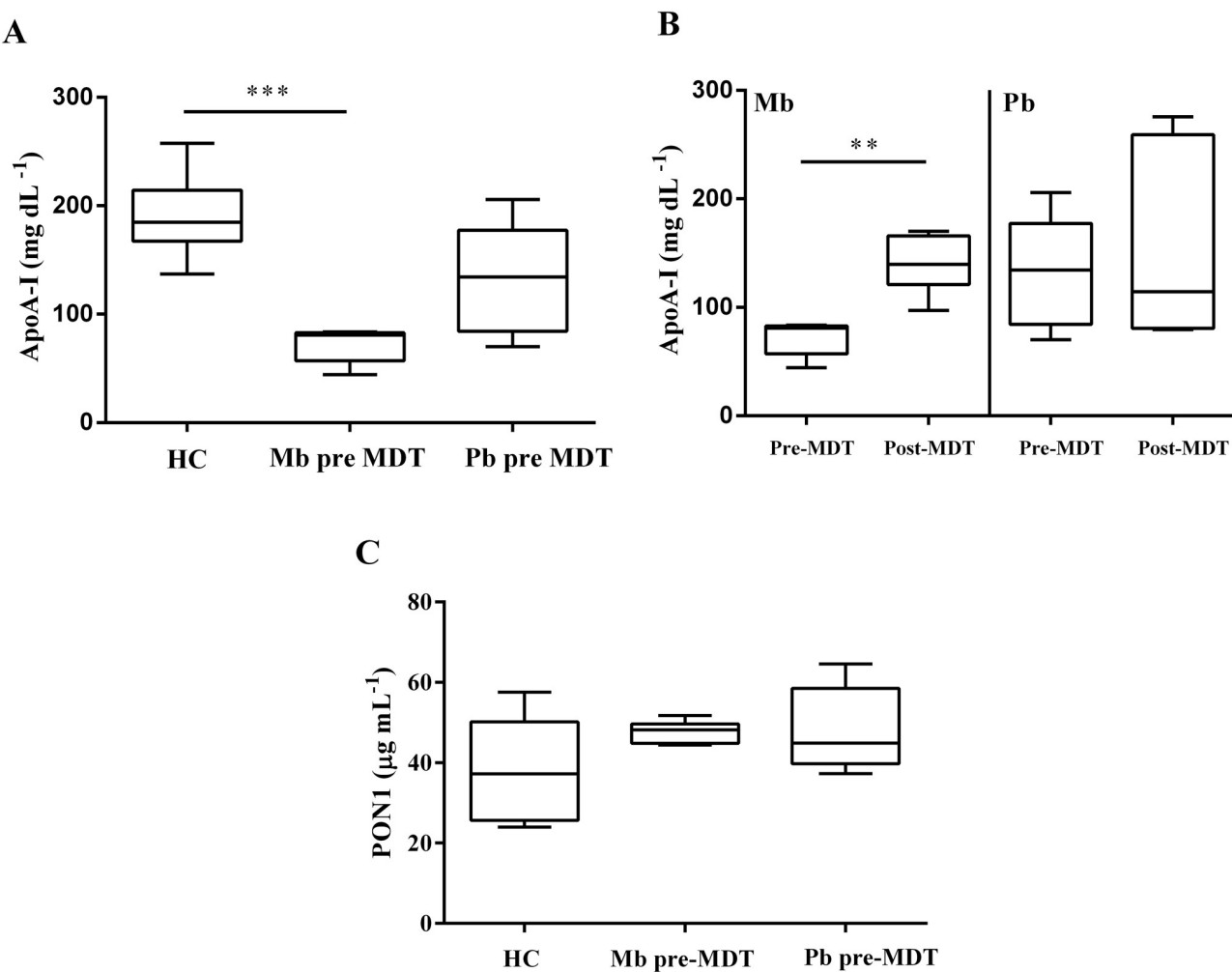

**Fig 5. Plasma levels of ApoA-I are drastically reduced in multibacillary leprosy patients.** Box-plots represent levels of ApoA-I in the plasma (A) of HC (n = 8), Mb pre-MDT (n = 5), and Pb pre-MDT (n = 6). (B) ApoA-I in the plasma of Mb pre-MDT (n = 5), Mb post-MDT (n = 6), Pb pre-MDT (n = 6), and Pb post-MDT (n = 6). (C) Plasma PON1 levels from HC (n = 7), Mb pre-MDT (n = 6), and Pb pre-MDT (n = 5). determined by EIA. Median and min-max values are indicated by lines. (A) and (C) HC, Mb pre-MDT and Pb-MDT comparisons were evaluated with Kruskal–Wallis nonparametric and Dunn's tests. (B) Mb and Pb patients were evaluated with the Mann Whitney non-parametric test. $^{**}p \leq 0.001$, $^{***}p \leq 0.0001$. The absence of $p$-value indicates non-significant differences.

the next step focused on an *in vitro* exposition of hepatic cells to *M. leprae* bacilli to check ApoA-I expression by immunoblotting. It was observed that *M. leprae* was able to associate with HepG2 cells (Fig 6D). Moreover, this association was able to decrease ApoA-I expression in a dose-dependent manner (Fig 6E). When comparing the unstimulated control with *M. leprae*-treated hepatic cells, it is possible to see an average reduction of 50 to 70% on Apo-I expression at MOI of 25:1 and 50:1.

## Discussion

In Mb leprosy, *M. leprae* modulates host lipid metabolism both in infected cells and systemically to evade the immune system and facilitate its survival [8, 9, 12]. Previous data have shown an accumulation of cholesterol and other host-derived lipids inside *M. leprae*-infected macrophages and Schwann cells resulting in a foamy phenotype [6–8]. However, the molecular basis

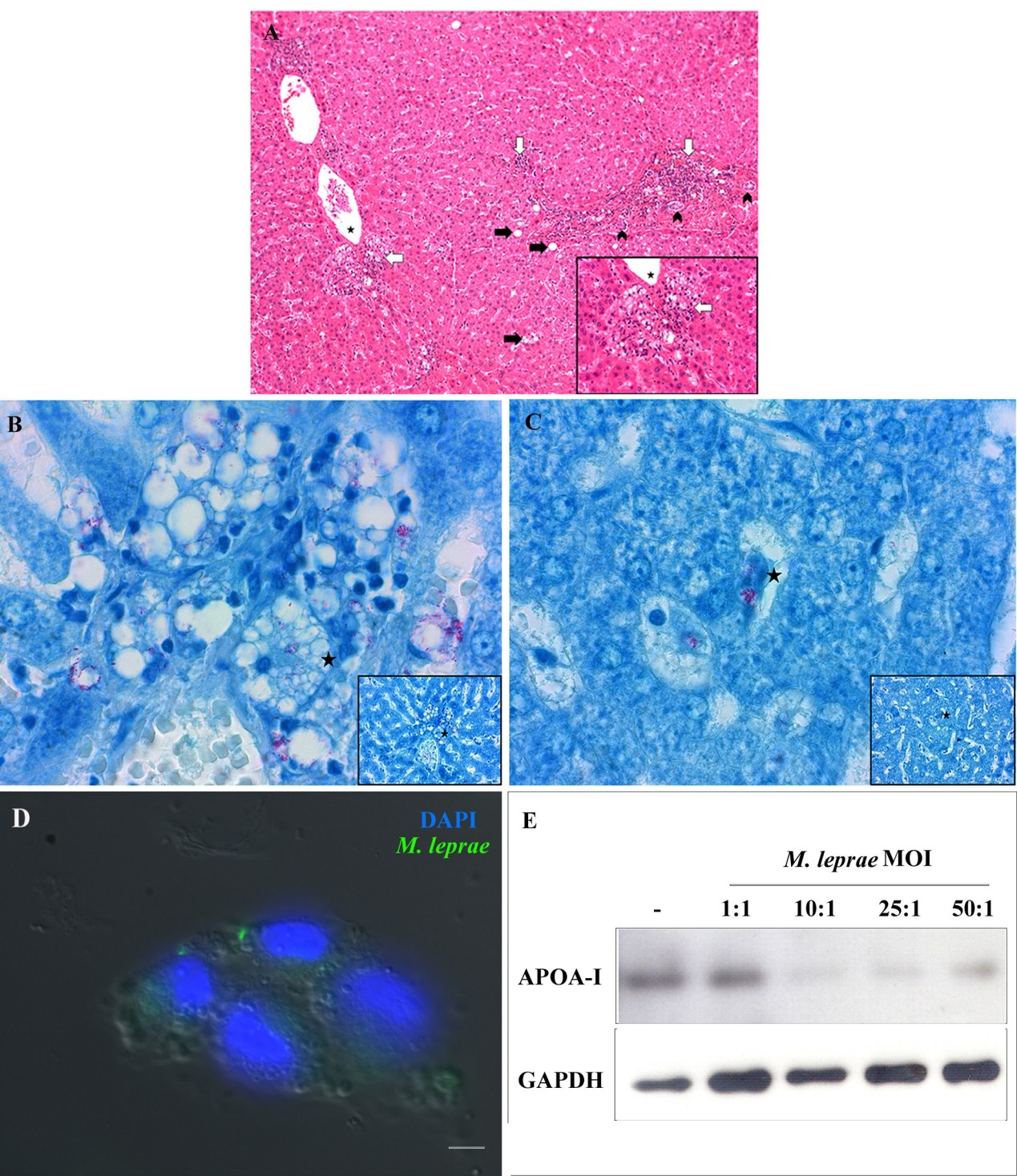

**Fig 6.** *M. leprae* **was detected in the liver of a multibacillary patient and can decrease ApoA-I expression in HepG2 cells.** (A, B, C) Photomicrography of a section from a Mb patient liver autopsy. (A) H&E-stained hepatic parenchyma with vascular congestion (stars) steatosis (black arrows). Extended portal spaces (arrowheads). Inflammatory infiltrate (white arrows); Magnification: 400x, inset 1000x. Fite-Faraco stained Virchow's cells with granular AFB (stars) in the inflammatory infiltrate (B) and in hepatic sinusoids (C); Magnification: 1000x, inset 400x. (D) Immunofluorescence image of HepG2 cells infected with *M. leprae* labeled in green (PKH) and nuclei in blue with DAPI; Scale bar, 5 μm. (E) ApoA-I immunoblotting from HepG2 treated with live *M. leprae* in different MOIs (1:1, 10:1, 25:1 and 50:1). (-) = Unstimulated. A representative experiment of four separate experiments is shown.

of this phenomenon is not fully understood. Findings from Cruz et al. [9] suggested that HDL from leprosy patients is dysfunctional, showing a lower capacity to act in the removal of oxidized lipids and to perform the reverse transport of cholesterol from the peripheral tissues, which are considered major functions of this lipoprotein in lipid metabolism homeostasis. Since the chemical composition of HDL influences its function [54], in the present study the lipid profile and protein levels of HDL from Mb patients were investigated to verify the influence of these factors on its defective phenotype. Our findings suggest that HDL from Mb has an altered lipid composition and decreased ApoA-I, which are partially reverted by MDT treatment. Moreover, our data suggest that *M. leprae* can directly affect ApoA-I biosynthesis in the liver since hepatic cell cultures treated with live bacteria showed lower levels of ApoA-I. It was also confirmed that Mb HDL presents impaired reverse cholesterol capacity. Additional assays were performed, which indicated lower antioxidant and anti-inflammatory capacities assessed by its decreased ability to suppress ROS propagation, and IL-6 and MCP-I production in endothelial cells. Most of the HDL's composition parameters evaluated were at least partially recovered after MDT conclusion, establishing a direct link between the observed alterations and active multibacillary leprosy.

Initially, it was observed that Mb pre-MDT patients (n = 8) compared to healthy controls (n = 12) presented reduced HDL-cholesterol concentrations with values varying between 21–39, min-max mg/dL. A worrying fact, since the Brazilian Longitudinal Multicentric Study of Adult Health (ELSA Brazil) [55] classified as desirable levels of HDL-cholesterol ≥40 mg/dL for men and ≥50 mg/dL for women. These data are in agreement with recent findings from Silva et al (2018) and Negera et al (2018), which also observed lower levels of HDL-cholesterol in Mb patients compared to Pb and HC [56, 57]. Other studies on serum lipids in leprosy also described lower levels of HDL-cholesterol in Mb patients [58, 59]. However, older studies reported higher HDL-cholesterol levels in Mb patients [60–63], probably due to different analytical methodologies and populations. Low levels of plasma HDL are considered as a risk factor in the development of diseases that have as one of the main features the lipid accumulation in tissues due to an inefficient lipid efflux activity [64] like atherosclerosis [65]. There are studies reporting autopsy findings in leprosy (both paucibacillary and multibacillary) where atherosclerosis and plaque calcification were observed in all cases [66, 67], whereas one study reported a low incidence of atherosclerosis in leprosy cases [68]. A Brazilian study observed atherosclerosis in 193 from 209 autopsies in leprosy patients [69]. Considering the more recent reports describing lower levels of HDL-cholesterol, as well as the presence of atherosclerosis on autopsies from leprosy patients, the relation between leprosy and atherosclerosis should be studied in more depth.

Beyond HDL-cholesterol data, UPLC-MS-based lipidomics data suggested that the lipid composition of HDL was altered in Mb pre-MDT patients. Considering that lipids were detected as "compound" groups, database search allowed the annotation of the compound POVPC on C273, in both protonated and sodiated molecular ions. POVPC (1-palmitoil-2-(5-oxovaleroil)-*sn*-glcero-3-phosphatidylcholine) was found in lower levels in Mb HDL, mainly when compared to HDL from Pb pre-MDT. POVPC is an oxidized form of 1-palmitoyl-2-arachidonoyl-*sn*-phosphatidylcholine (PAPC), commonly found in atherosclerotic lesions and oxidized LDL [70]. Indeed, POVPC has also been shown to accumulate in leprosy lesions [9]. Altogether, the lower levels of HDL-cholesterol and apparently of POVPC reinforce the idea that HDL from Mb patients is dysfunctional, being unable to mediate the removal of cholesterol and oxidized phospholipids from peripheral tissues. Furthermore, comparative functional assays performed *in vitro* confirmed that Mb HDL exhibits a lower ability to promote the cholesterol efflux in comparison with HC's HDL, agreeing with previous studies [9].

Cholesterol efflux is attributed to the HDL-associated protein ApoA-I [71]. ApoA-I occupies 70% of HDL structure and performs most of its functions (reviewed in [43]). A drastic decrease in ApoA-I levels was observed on Mb pre-MDT patients, correlating with the lower capacity of HDL to mediate cholesterol efflux in Mb patients, which seems to be also lowered in plasma of Mb leprosy patients when compared to HC and Pb patients.

Further functional analyzes showed that HDL from Mb pre-MDT has also a lower antioxidant activity against the oxidative stress promoted by $H_2O_2$ when compared to HDL from HC. This is also possibly linked with the lower amount of ApoA-I, since previous works showed that the prevention of oxidative stress by HDL is mainly performed by ApoA-I and PON-1 [72, 73] and data showed no differences on PON-1 plasma levels between patients and HC. When evaluating the inflammatory activity of HDL, it was revealed that this lipoprotein from Mb pre-MDT patients increased the expression of IL-6 upon LPS stimulation compared to HDL from HC, but it was unable to modulate the basal expression of MCP-1 on endothelial cells. ApoA-I can prevent the release of pro-inflammatory cytokines by inhibiting the binding of LPS with TLR4 [74–76], and by reducing the activation of NF-κB (nuclear transcription factor-κB) through its interaction with SR-B1 (scavenger receptor class B type 1) [77, 78]. Together, the data presented here indicate that low ApoA-I content on HDL structure is responsible for the downregulation of its functions in Mb pre-MDT patients.

Considering the important role of ApoA-I on dysfunctional HDL seen in Mb pre-MDT patients, the next step focused on elucidating the source of plasma ApoA-I deficit. Normally, plasma alterations of ApoA-I are caused by an imbalance between its synthesis and catabolism [79–81]. In acute inflammations triggered by infections, high levels of serum amyloid A (SAA) can catabolize ApoA-I in HDL, which is commonly seen on cases of sepsis [81, 82] or erythema nodosum leprosum (ENL), where high levels of SAA are associated to low levels of HDL-cholesterol [58]. Also, it has been shown that ApoA-I can be covalently modified by reactive carbonyls, namely malondialdehyde (MDA) [83]. MDA is a peroxidation product of polyunsaturated fatty acids (PUFA) produced during oxidative stress [84, 85] and can be associated with low HDL-cholesterol and ApoA-I levels in metabolic syndromes where elevated oxidative stress is observed [86]. MDA-modified HDL is susceptible to internalization by scavenger receptors, leading to degradation in lysosomes of sinusoidal liver cells [87]. Recent findings indicate that Mb patients have high plasma levels of MDA compared to Pb patients [88–90]. These data are a strong indicative that Mb patients could have impaired activity of ApoA-I due to its damage by MDA.

On the other hand, by an anabolism perspective, modulation of ApoA-I levels in leprosy may be correlated to a hepatic involvement, as *APOA1* gene is mainly expressed by hepatocytes. Histological examination of Mb patient's liver indicated alterations on vasculature and morphology of hepatic parenchyma with granulation tissue containing bacilli. Focal liver inflammatory infiltration is common in Mb patients [52, 91, 92] and in some cases, hepatocytes with high fatty content (steatosis) and vacuolar degenerative profile can be observed [92]. Hepatocyte damage is commonly triggered during inflammations promoted by chronic exposition to cytokines like TNF (tumor necrosis factor) [93, 94]. TNF is a typical pro-inflammatory cytokine released by Kupffer cells in hepatic granulomas induced by BCG (*M. bovis* Calmette-Guérin) and *M. tuberculosis*, to keep their morphological structure in the hepatic environment [95, 96]. High levels of TNF and interleukin-1β in hepatic environment can negatively regulate *APOA1* gene expression via nuclear receptors LXRs (liver X receptor), FXRs (farnesoid X receptor) and PPARα (peroxisome proliferator-activated receptor α) in hepatocytes [97–99], and maybe leading the modulation on ApoA-I expression in leprosy patients.

Modulation of ApoA-I expression in liver by *M. leprae* seems to be complex and should be approached by different hypothesis. The chronic disease observed on leprosy patients creates

an environment of constant inflammation and, because of the presence of *M. leprae* in the blood of multibacillary patients [100], maybe exposing hepatic cells directly to *M. leprae*. To test this hypothesis, the direct exposition of hepatic cells (HepG2) cells *in vitro* to live *M. leprae* was performed, and a down-regulation on ApoA-I synthesis was observed. Normally hepatic cells own all the machinery necessary to act as an activated innate immune cell and promote the killing of invasive bacteria by secreting pro-inflammatory mediators like IL-6, IL-22, IL-1β and TNF [101], which would explain the modulation of ApoA-I expression. Taking into account the systemic balance between catabolism and synthesis of ApoA-I in leprosy, the present work opens a new perspective about ApoA-I modulation and hepatic lipoprotein metabolism impairment in leprosy, which will be the aim of future studies.

A major limitation of our study concerns the low number of cases analyzed, mainly due to the complex process to purify HDL samples. However, despite the small sample size, our data strongly suggest lower levels and dysfunctional HDL in Mb patients, which points to the necessity to explore the pathophysiological consequences of HDL dysfunctionality in leprosy in the future. Of note, the alignment between this study and the literature may direct a new perspective on the use of medicines which can reverse HDL dysfunctionality, favoring patient recovery, together with better efficiency of MDT.

## Supporting information

**S1 Fig. Principal component analysis of the lipidomics profile of purified HDL obtained from leprosy patients and healthy control individuals.** Raw UPLC-MS data collected in positive ionization mode were processed and analyzed by XCMS, followed by normalization through the quantile method. Subsequently, the molecular features (MFs) were grouped into spectra through RAMClustR. In this approach, each spectrum represents a "compound" with adducts, isotopes and monoisotopic mass. The intensities of 1260 "compounds" were used to perform a principal component analysis. HC (n = 6, black dot), Pb pre-MDT (n = 5, grey dot), Pb post-MDT (n = 4, green dot), Mb pre-MDT n = 4, blue dot), Mb post-MDT (n = 5, red dot).
(TIF)

**S2 Fig. Principal component analysis indicates that the lipidomics profile of HDL from Mb patients differ from Pb patients and HC.** Raw UPLC-MS data collected in positive ionization mode were processed and analyzed by XCMS, followed by normalization through the quantile method. Subsequently, the molecular features (MFs) were grouped into spectra through RAMClustR. In this approach, each spectrum represents a compound group (named "compound") with adducts, isotopes and monoisotopic mass. The intensities of 1260 "compounds" were used to perform a principal component analysis. (a) HC (n = 6, black dot), Pb pre-MDT (n = 5, grey dot) and Pb post-MDT (n = 4, green dot). (b) HC, Pb post-MDT and Mb post-MDT patients (n = 5 red dot). (c) HC individuals, Mb pre-MDT patients (n = 4, blue dot) and Pb patients. (d) HC, Mb patients and Mb post-MDT patients.
(TIF)

**S3 Fig. HDL from multibacillary leprosy patients display altered functions.** (A) Inhibition of ROS (reactive oxygen species) by HDL purified from leprosy patients was measured by flow cytometry. Data are expressed by the median fluorescence intensity (MFI). Mb pre-MDT (n = 5), Mb post-MDT (n = 5), Pb pre-MDT (n = 5) and Pb post-MDT (n = 6). (B) Expression of monocyte chemoattractant protein-1 (MCP-1) modulated by HDL from leprosy patients in HCAEC cells. Mb pre-MDT (n = 5), Mb post-MDT (n = 5), Pb pre-MDT (n = 6) and Pb post-MDT (n = 4). (D) Pb and Mb HDL-cholesterol efflux from THP-1 differentiated macrophages.

Mb pre-MDT (n = 5), Mb post-MDT (n = 5), Pb pre-MDT (n = 4) and Pb post-MDT (n = 6). Median and min-max values are indicated by lines. Group comparisons were evaluated using Mann-Whitney test.
(TIF)

**S1 Table. Number of "compounds" with $\log_2$FC and $p<0.05$ in the comparisons made in the current study.**
(TIF)

**S2 Table. Statistical comparison of the spectral entities ("compounds") with their respective $\log_2$ fold change and $p$ values.**
(PDF)

**S3 Table. Putative identification of the "compounds" with $\log_2$FC$\geq$1.0 and $p<0.05$.**
(PDF)

## Acknowledgments

We are grateful to Andréa F. Belone and Karyn L. Hamilton for providing microscopic autopsy liver sections and HCAEC, respectively. We thank Corey Broeckling for UPLC-MS, Flávio Alves Lara for microscopy and HPLC and Mariana Hacker for statistical analysis support.

## Author Contributions

**Conceptualization:** Robertha Mariana R. Lemes, Maria Cristina V. Pessolani, Cristiana S. de Macedo.

**Formal analysis:** Robertha Mariana R. Lemes, Carlos Adriano de M. e Silva.

**Funding acquisition:** Maria Cristina V. Pessolani.

**Investigation:** Robertha Mariana R. Lemes, Maria Renata S. Nogueira.

**Methodology:** Robertha Mariana R. Lemes, Maria Ângela de M. Marques, Georgia C. Atella, Maria Renata S. Nogueira, Prithwiraj De, Delphi Chatterjee, Maria Cristina V. Pessolani, Cristiana S. de Macedo.

**Project administration:** Maria Cristina V. Pessolani, Cristiana S. de Macedo.

**Resources:** Georgia C. Atella, José Augusto da C. Nery, Patricia S. Rosa, Cléverson T. Soares, Delphi Chatterjee, Maria Cristina V. Pessolani.

**Supervision:** Maria Cristina V. Pessolani, Cristiana S. de Macedo.

**Writing – original draft:** Robertha Mariana R. Lemes, Carlos Adriano de M. e Silva, Maria Cristina V. Pessolani, Cristiana S. de Macedo.

**Writing – review & editing:** Carlos Adriano de M. e Silva, Prithwiraj De, Delphi Chatterjee, Maria Cristina V. Pessolani, Cristiana S. de Macedo.

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
