## [Decision Letter · Decision Letter 0]

24 Dec 2019

Dear Dr. de Macedo:

Thank you very much for submitting your manuscript "Altered composition and functional profile of high-density lipoprotein in leprosy patients" (PNTD-D-19-00946) for review by PLOS Neglected Tropical Diseases. Your manuscript was fully evaluated at the editorial level and by independent peer reviewers. The reviewers appreciated the attention to an important topic but identified some aspects of the manuscript that should be improved.

We therefore ask you to modify the manuscript according to the review recommendations before we can consider your manuscript for acceptance. Your revisions should address the specific points made by each reviewer.

(1) A letter containing a detailed list of your responses to the review comments and a description of the changes you have made in the manuscript.

(2) Two versions of the manuscript: one with either highlights or tracked changes denoting where the text has been changed (uploaded as a "Revised Article with Changes Highlighted" file ); the other a clean version (uploaded as the article file).

(3) If available, a striking still image (a new image if one is available or an existing one from within your manuscript). If your manuscript is accepted for publication, this image may be featured on our website. Images should ideally be high resolution, eye-catching, single panel images; where one is available, please use 'add file' at the time of resubmission and select 'striking image' as the file type. 

Please provide a short caption, including credits, uploaded as a separate "Other" file. If your image is from someone other than yourself, please ensure that the artist has read and agreed to the terms and conditions of the Creative Commons Attribution License at http://journals.plos.org/plosntds/s/content-license (NOTE: we cannot publish copyrighted images). 

(4) Appropriate Figure Files 

Please remove all name and figure # text from your figure files upon submitting your revision. Please also take this time to check that your figures are of high resolution, which will improve both the editorial review process and help expedite your manuscript's publication should it be accepted. Please note that figures must have been originally created at 300dpi or higher. Do not manually increase the resolution of your files. For instructions on how to properly obtain high quality images, please review our Figure Guidelines, with examples at: http://journals.plos.org/plosntds/s/figures

While revising your submission, please upload your figure files to the Preflight Analysis and Conversion Engine (PACE) digital diagnostic tool, https://pacev2.apexcovantage.com/ PACE helps ensure that figures meet PLOS requirements. To use PACE, you must first register as a user. Then, login and navigate to the UPLOAD tab, where you will find detailed instructions on how to use the tool. If you encounter any issues or have any questions when using PACE, please email us at figures@plos.org.

We hope to receive your revised manuscript by Feb 22 2020 11:59PM. If you anticipate any delay in its return, we ask that you let us know the expected resubmission date by replying to this email.

To submit your revised files, please log in to https://www.editorialmanager.com/pntd/

Sincerely,

Richard Odame Phillips

Deputy Editor

Richard Phillips

Deputy Editor

Reviewer's Responses to Questions

**Key Review Criteria Required for Acceptance?**

**Methods**

-Are the objectives of the study clearly articulated with a clear testable hypothesis stated?

-Is the study design appropriate to address the stated objectives?

-Is the population clearly described and appropriate for the hypothesis being tested?

-Is the sample size sufficient to ensure adequate power to address the hypothesis being tested?

-Were correct statistical analysis used to support conclusions?

-Are there concerns about ethical or regulatory requirements being met?

Reviewer #1: low number of patients but proof of principle study

the hypothesis was not clearly articulated with a testable

no statistical analysis

ethical fulfilled

Reviewer #2: The definitions of healthy controls and post-MDT participants were not clearly described. It was also unclear how the sample size was determined.

**Results**

-Does the analysis presented match the analysis plan?

-Are the results clearly and completely presented?

-Are the figures (Tables, Images) of sufficient quality for clarity?

Reviewer #1: yes, clear results and completely presented

figures are OK

Reviewer #2: Yes

**Conclusions**

-Are the conclusions supported by the data presented?

-Are the limitations of analysis clearly described?

-Do the authors discuss how these data can be helpful to advance our understanding of the topic under study?

-Is public health relevance addressed?

Reviewer #1: conclusions are supported by the data except the low number of cases as written above. but this can be considered as a fundamnetal research study showing what to search for in the future.

Limitations could be more described

public relevance could be addressed

Reviewer #2: Yes

**Editorial and Data Presentation Modifications?**

Reviewer #1: Minor comments

Shortening the material and methods since it is more than 8 pages. Most of the methods were already published and can be briefly described mentioning the differences done in this study.

Lines 519-521: revise the sentence and add the number of patients involved in this Brasilian cohort

Reviewer #2: - Line 105-109 in page 4 should be moved to the Methods section

- Line 107-109 in page 4 “Moreover” would be better removed 

- Line 122-124 in page 4 needs to be rephrased

- Line 125-126 in page 5 should be removed as it is also stated in the ethics statement section

- Line 131 in page 5 “Participants” is preferred over “subjects”

- Line 213 in page 8 should be “…and incubated for more than one hour”

- Line 263 in page 10 should be “Prior to the extraction…”

- Line 285 in page 11 should be “..held constant throughout the experiments”

- Line 483 in page 19 should be “…, it is possible to see...”

- Line 536 in page 20 should be “…, as well as the presence of atherosclerosis…”

**Summary and General Comments**

Reviewer #1: The authors investigated the lipid metabolism of patients with leprosy. 

The methods seem adequate and well described. The paper is well written.

Leprosy is still a disease lacking knowledge on the host reaction to the infection, especially at the molecular level, and also on the metabolic characteristics of patients developing the disease. In this study, a part of the lipid metabolism, concentration of HDL and specifically ApoA-1, and its functionality, lipid fractions and HDL functionality were explored with regard to patients with either a paucibacillary or multibacillary disease, or healthy controls. Since decrease in HDL can be seen as a risk factor rather than the consequence of leprosy, the authors infected hepathic cells with M. leprae and showed that it also induced a decrease in the expression of ApoA-1. This is explained by ApoA-1 inability to remove cholesterol from peripheral tissues, which is often observed in MB cases.

Lipid metabolism may be a major issue for leprosy treatment and management in the future and increase in the knowledge is necessary. This paper brings this. 

Major comments

The number of patients with MB before MDT is low, i.e. the patients considered as the “cases” are only 6 compared to a similar number of controls distributed as healthy controls, PB patients, and moreover MB after MDT and PB after MDT. Due to low numbers, it could have been more relevant to distinguish three studies, one with MB compared to HC, one comparing MB to PB and one comparing before and after MDT. Overall all the groups have low number of cases or controls and consequently what was observed is of low statistical values. How did the authors calculate the statistical differences?

Moreover, the patients after MDT are not the same as those before MDT. So interpatient comparison is difficult to assess.

Reviewer #2: The manuscript “Altered composition and functional profile of high-density lipoprotein in leprosy patients” describes a comprehensive study that evaluates the association of HDL metabolism in leprosy. Interestingly, the authors have successfully demonstrated an altered HDL composition and its function in multibacillary leprosy patients. This manuscript brings interesting and important information that could contributes to the development of leprosy management. The authors deserve appreciation for their effort in conducting such study. 

Overall, the manuscript is well-written and the study population was appropriate for the working hypothesis. Nonetheless, the definitions of healthy controls and post-MDT participants need to be more explained. The authors would have possibly mentioned the definition of post-MDT group in line 123-125. However, the sentence is not clear enough and therefore needs to be rephrased. Furthermore, case-control comparisons in leprosy research are likely biased when hospital or care unit controls consequently do not represent the exposure experience of the true source population. This is a care unit-based study where all the participants were recruited on a volunteer basis and choosing suitable controls for such study is often difficult. Therefore, it would be better if the authors can provide more information on the selection of the healthy controls. While the authors can already demonstrate the evidence for the hypothesis being tested with a small number of participants, it would also be better if the authors can provide more information on how the total sample size of 39 was determined.

PLOS authors have the option to publish the peer review history of their article (what does this mean?). If published, this will include your full peer review and any attached files.

Reviewer #1: Yes: Emmanuelle Cambau

Reviewer #2: No

---

## [Editor Report · Decision Letter 1]

12 Feb 2020

Dear Dr. de Macedo,

We are pleased to inform you that your manuscript 'Altered composition and functional profile of high-density lipoprotein in leprosy patients' has been provisionally accepted for publication in PLOS Neglected Tropical Diseases.

Before your manuscript can be formally accepted you will need to complete some formatting changes, which you will receive in a follow up email. A member of our team will be in touch within two working days with a set of requests.

Best regards,

Richard Odame Phillips

Deputy Editor

Richard Phillips

Deputy Editor

---

## [Editor Report · Acceptance letter]

11 Mar 2020

Dear Dr. de Macedo,

We are delighted to inform you that your manuscript, "Altered composition and functional profile of high-density lipoprotein in leprosy patients," has been formally accepted for publication in PLOS Neglected Tropical Diseases.

Best regards,

Serap Aksoy

Editor-in-Chief

Shaden Kamhawi

Editor-in-Chief
